# Studies in Rats of Combined Muscle and Liver Perfusion and of Muscle Extract Indicate That Contractions Release a Muscle Hormone Directly Enhancing Hepatic Glycogenolysis

**DOI:** 10.3390/jpm12050837

**Published:** 2022-05-20

**Authors:** Xiao X. Han, Jens J. Holst, Henrik Galbo

**Affiliations:** 1Department of Rheumatology, Institute of Inflammation Research, Rigshospitalet, University of Copenhagen, Tagensvej 20, 2200 Copenhagen, Denmark; xiaoxiahan@ymail.com; 2Department of Biomedical Sciences, NovoNordisk Foundation Center for Basic Metabolic Research, Faculty of Health Sciences, University of Copenhagen, 2200 Copenhagen, Denmark; jjholst@sund.ku.dk

**Keywords:** glucose, liver, muscle contraction, exercise, glucose turnover, myokine

## Abstract

Background: Established neuroendocrine signals do not sufficiently account for the exercise-induced increase in glucose production. Using an innovative, yet classical cross-circulation procedure, we studied whether contracting muscle produces a factor that directly stimulates hepatic glycogenolysis. Methods: Isolated rat hindquarters were perfused in series with isolated livers. Results: Stimulation of the sciatic nerve of one or both legs resulted in an increase in force, which rapidly waned. During one-legged contractions, hepatic glucose production increased initially (from −0.9 ± 0.5 (mean ± SE) to 3.3 ± 0.7 µmol/min, *p* < 0.05). The peak did not differ significantly from that seen after 20 nM of epinephrine (5.1 ± 1.2 µmol/min, *p* > 0.05). In response to two-legged contractions, the increase in hepatic glucose production (to 5.4 ± 1.3 µmol/min) was higher (*p* < 0.05) and lasted longer than that seen during one-legged contractions. During contractions, peak hepatic glucose output exceeded concomitant hepatic lactate uptake (*p* < 0.05), and glucose output decreased to basal levels, while lactate uptake rose to a plateau. Furthermore, in separate experiments an increase in lactate supply to isolated perfused livers increased lactate uptake, but not glucose output. In intact rats, intra-arterial injection of extract made from mixed leg muscle elicited a prolonged increase (*p* < 0.05) in plasma glucose concentration (from 5.2 ± 0.1 mM to 8.3 ± 1.5 mM). In perfused livers, muscle extract increased glucose output dose dependently. Fractionation by chromatography of the extract showed that the active substance had a MW below 2000. Conclusion: This study provides evidence that contracting skeletal muscle may produce a hormone with a MW below 2000, which enhances hepatic glycogenolysis according to energy needs. Further chemical characterization is warranted.

## 1. Introduction

Glucose is an important fuel for working muscle [1,2,3]. In man, muscular glucose uptake accounts for 89% of carbohydrate combustion and 30% of overall metabolism in light exercise, while in heavy exercise, the corresponding figures are 63% and 41%, respectively [1]. Increases in glucose uptake in working muscle are accompanied by increases in hepatic glucose production predominantly reflecting glycogenolysis. In general, glucose production is well matched to glucose utilization and the plasma glucose concentration only changes a little during exercise. Concerning the regulation of hepatic glucose production, it is thought that both feedforward mechanisms—which are intimately related to motor activity—and feedback mechanisms—which are intimately related to glucose needs—are involved [4,5,6].

This view is based on studies of glucose production during exercise and of the neuroendocrine signals putatively involved in its regulation. It is believed that from the onset of exercise, nervous impulses from motor centers in the brain and from working muscles stimulate neuroendocrine centers to elicit a work-rate-dependent increase in sympathetic nerve activity and in release of GH (growth hormone) and ACTH (adrenocorticotropic hormone) from the pituitary gland. These responses, then, control the changes in secretion of subordinate glands: sympathetic activity enhances the secretion of epinephrine and, in some lower species, glucagon, whereas insulin secretion is depressed. ACTH stimulates cortisol secretion. The fast nervous mechanisms related to motor activity operate throughout exercise, but the hormonal responses may be gradually intensified due to feedback from error signals, among which a decrease in glucose availability is the most important metabolic one. In humans, an increase in plasma glucagon is predominantly determined by a decrease in plasma glucose concentration and is not seen until after one hour of moderate exercise [4].

However, the signals directly triggering an increase in hepatic glucose production in exercise are far from fully elucidated. A decrease in plasma insulin concentration is essential [5,6]. Furthermore, in rats and dogs, increases in epinephrine and glucagon concentrations probably also play a role [7,8,9,10]. Regarding man, however, the well-known neuroendocrine signals do not sufficiently account for the exercise-induced increase in glucose production [2,11,12]. This is illustrated by the fact that in experiments in which the celiac ganglion, and accordingly liver nerve activity and epinephrine secretion, was blocked, and glucagon and insulin concentrations were clamped at basal levels, glucose production nevertheless increased from the onset of exercise [13]. Glucose production reached a plateau, while a continuous decrease in plasma glucose was observed. The finding suggests the existence of an additional liver-stimulating factor more closely related to motor activity than to glucose needs [13].

An attractive possibility would be that contracting muscle produces a factor that directly stimulates hepatic glucose production [5]. Work-intensity-dependent secretion of a glycogenolytic hormone by muscle would appear appropriate, making tight coupling possible between the need for and delivery of glucose to muscle. Since we first introduced this idea [5,14], skeletal muscle has, in fact, been shown to produce and secrete hundreds of putative peptide and protein factors (termed “myokines”) [15,16,17,18]. However, the physiological roles of these substances are still far from fully clarified, and yet none have been shown to stimulate liver glucose production [15,16,17,19,20].

It is agreed that there is a need for direct studies of the physiological actions of the tentative myokines [19,21]. So, in the present study we used an innovative, yet classical cross-circulation procedure to elucidate whether contracting muscle in fact produces a factor which can stimulate hepatic glucose production. Rat hindquarter (essentially reflecting muscle metabolism [22]) and liver were isolated and perfused in series, and muscle was electrically stimulated. Furthermore, crude and fractionated muscle extracts were injected in intact rats or isolated livers, and glucose responses were studied. The presented findings indicate that contracting skeletal muscle may produce a circulating factor with a MW below 2000, which enhances hepatic glycogenolysis.

## 2. Materials and Methods

### 2.1. Hindquarter and Liver Perfusion in Series

A short explanation of the idea behind the experimental setup: Muscle (rat hindquarter) was perfused in vitro from the aorta, and the venous outflow served as perfusate for an isolated liver. Accordingly, factors released from muscle were carried by the perfusate directly to the liver. Muscle could be electrically stimulated and any effects of released factors on hepatic glucose output could be measured from perfusate flow and perfusate glucose concentration differences across the liver.

Two fed Wistar rats, weighing about 250 g, were used in each combined hindquarter and liver perfusion. One rat was anesthetized with intraperitoneal injection of pentobarbital sodium (5 mg·100 g body wt^−1^) and then surgically prepared for hindquarter perfusion as described in [22]. One thousand IU heparin were injected into the inferior vena cava, and catheters were inserted into the aorta and inferior vena cava. The rat was killed by intracardial injection of 0.5 mL pentobarbital sodium (50 mg/mL) and put on the perfusion apparatus (Appendix A). The initial perfusate volume was 700 mL containing Krebs–Henseleit solution, 6 mM glucose, bovine erythrocytes (obtained in a local abattoir 1–2 days before perfusion) at a hematocrit of 30%, 0.15 mM pyruvate, and 4% bovine serum albumin (BSA) dialyzed (cutoff at 10–15 K Da) for 48 h with Krebs–Henseleit solution. Perfusate was adjusted with NaOH to pH 7.45. To reduce the lactate production of the erythrocytes, the perfusate reservoir was kept on ice throughout the perfusion. Before entering the hindquarter, the perfusate was gassed with 98% O_2_ and 2% CO_2_ (V:V) and heated to 37 °C. Hindquarters were initially perfused with non-recirculated perfusate at a flow of 7–8 mL/min.

When the hindquarter perfusion was started, the other rat was anesthetized and heparinized. A catheter was inserted into its portal vein and connected to the venous catheter from the perfused hindquarter (Appendix A). To allow liver perfusion, a hole was immediately made in the lower part of the inferior vena cava. The liver artery and bile duct were ligated, and the portal venous catheter was secured by strings. Then, the chest was opened by sternotomy and a catheter was led through the right atrium to a position in the inferior vena cava just above the diaphragm. The vein was closed around the catheter by strings. Then, the inferior vena cava was tied off beneath the kidney veins and just above the artificial hole. The rat was killed by injection of pentobarbital sodium into the left ventricle of the heart. Liver isolation lasted about 10 min.

When muscle contraction was performed, the knee joint of one or both legs was fixed with a steel pin beneath the tibiopatella ligament. A string tied around the Achilles tendon was connected to a Harvard Instrument Isometric Muscle Transducer (Harvard Instrument, Millis, MA, USA), which was linked to a dual-channel Astro-Med recorder (Atlan. Tol Industries, West Warwick, RI, USA). The sciatic nerve was stimulated by a hook electrode to produce repeated tetanic contraction. Electrical stimulation was given as 200 ms trains of 100 Hz, which were delivered every second at supramaximal voltage (30 V), and with each impulse in the train lasting 0.1 ms. The perfusate flow was increased just before stimulation (see below), and the muscle length was adjusted during initial contractions to achieve the maximum increase in tension during contractions.

The rats were pre-perfused for 40 min with non-recirculated perfusate at a flow of 9 mL·min^−1^. To allow recirculation of the medium, the catheter in the inferior vena cava (above the diaphragm) was then connected to the perfusate reservoir. Then, 45 min (two-legged contraction experiments) and 65 min (one-legged contraction experiments) later, leg muscle was stimulated indirectly to contract for 60 min (both legs) or 85 min (one leg), respectively, at a flow of 15 mL/min. After one leg stimulation, flow was reduced to 9 mL/min, and epinephrine was added to the reservoir to give a final concentration of 20 nmol/L in cell-free perfusate. In separate experiments, perfusate flow was varied without muscle stimulation.

Perfusate samples from the hindquarter artery and vein and from the hepatic vein were drawn for measurements of glucose and lactate with 5–10 min intervals. Substrate balances were calculated from these measurements and perfusate flow. Perfusate gas tensions and pH were measured in perfusate sampled at rest, and at the 15th and 55th min of contractions. Biopsies of liver and of the superficial part of the gastrocnemius muscle (WG; consisting mainly of fast-twitch white fibers), the deep part of the medial head of the gastrocnemius muscle (RG; consisting mainly of fast-twitch red fibers), and the soleus muscle (Sol; consisting mainly of slow-twitch red fibers) [23] were freeze clamped for glycogen analysis at the end of contractions. In experiments with stimulation of both legs, perfusate was sampled from the hindquarter vein for epinephrine and norepinephrine measurements 20 min before and 5 and 55 min after the beginning of contractions.

### 2.2. Lactate Infusion in Isolated Liver

To elucidate the influence of lactate supply per se on hepatic glucose production, isolated livers (see above) were perfused with non-recirculated standard perfusate (hematocrit 30%) with lactic acid (Sigma, Saint Louis, MO, USA) added in concentrations corresponding to those in the portal vein in experiments with combined muscle and liver perfusions. The flow was 9 mL/min for 30 min with 1 mmol/L lactate in the cell-free perfusate. Then, the lactate concentration in the perfusate was raised to about 3 mmol/L and the flow to 15 mL/min, and perfusion was continued for a further 20 min. Concentrations of lactate and glucose in the portal and caval veins were measured every 5 min.

### 2.3. Muscle Extract Injection

Mixed rat leg muscles and epididymal fat pads (control) were cut out, washed with cold 0.9% NaCl, cut into pieces, and boiled in 0.9% NaCl for 10 min, the NaCl volume to wet tissue weight ratio being 10:1 (mL:g) [24]. The boiling regimen has previously been used to extract peptides from the gut [25]. In other experiments, the same tissues were extracted with cold sodium phosphate buffer (pH 7.2) and without boiling [24]. The tissues were homogenized and centrifuged, and the supernatant was collected and stored at −70 °C until injection. Responses to extracts made by the two procedures were similar and only data obtained with extracts made by boiling are presented.

For extract injection, rats weighing 280 g were anesthetized as described above, and the carotid artery was cannulated and kept patent with heparin. After 50 min of recovery, two arterial blood samples were drawn with a 10 min interval to give basal values. Then, 2 mL (3.6% of ECV) of tissue extracts, corresponding to 200 mg muscle or adipose tissue, or of 0.9% NaCl were injected into the carotid artery. Subsequently blood samples were collected every 10 min for 1 h in order to determine changes in glucose and lactate concentrations.

### 2.4. Muscle Extract Infusion in Isolated Liver

Extracts of 1.2 g (low concentration) or 3.6 g (high concentration) of muscle were prepared by boiling, filtered (Swinnex, Millipore, Merck, Burlington, MA, USA, pore size 0.45 μ), and freeze dried before resuspension in 4 mL of distilled water. Rat livers were isolated and perfused with non-recirculated, oxygenated, cell-free standard perfusate at a flow of 16 mL/min for 20 min. Subsequently, the perfusate (100 mL) was circulated for 15 min, before muscle extract was added to the perfusion reservoir for 30 min perfusion. Perfusate samples were drawn every 5 min from the portal and cava veins for glucose and lactate measurements.

Concentrations of epinephrine and norepinephrine were measured in muscle extracts as well as in cell-free perfusate from the hepatic vein after 10 and 25 min of perfusion with extracts.

### 2.5. Muscle Extract Fractionation

In order to characterize the molecular size of muscle extract substances capable of stimulating hepatic glucose production, gel filtration was carried out. Freeze-dried extract of 3.6 g muscle was dissolved in 6 mL of 0.5 M acetic acid and loaded on a K16/100 G50 Sephadex fine gel filtration column (Pharmacia, Uppsala, Sweden) equilibrated and eluted with 0.5 mol/L acetic acid at 4 °C. ^125^I-labeled albumin and ^22^NaCl were added in trace amounts for calibration of the column and quality control of the gel filtration. The flow rate was 2.6 mL/8 min and 100 fractions were collected. The 45 fractions between V_O_ (elution volume of ^125^I-albumin) and V_T_ (elution volume of ^22^NaCl) were pooled in groups of 9 and designated fractions I–V, fraction V representing the original fractions 60–68. The pooled fractions were freeze dried and stored at −20 °C. Upon resuspension in 4 mL of distilled water, fractions I–V were studied in separate liver perfusions, as described above for crude extracts.

### 2.6. Analyses

Concentrations of glucose and lactate were analyzed using an automatic glucose and lactate analyzer (YSI, Yellow Springs Instruments, YSI Incorporated, Yellow Springs, OH, USA). The concentrations of glycogen in both muscle and liver were measured as glucose, using hexokinase and fluorometry after acid hydrolysis, and expressed as μmol glucose·g wet wt^−1^ [26]. Catecholamine concentrations were determined by a single isotope radioenzymatic method [27] evaluated previously [28]. Perfusate gas tensions and pH were measured by an acid-base analyzer (ABL-30, Radiometer Copenhagen, Radiometer Medical ApS, Copenhagen, Denmark).

### 2.7. Statistics

Data are shown as mean ± SE. Effects of treatment or time were evaluated by Student’s paired *t*-test or one-way analysis of variance (ANOVA) with repeated measures as applicable. Differences revealed by ANOVA were located by Newman–Keuls post hoc test. If compatibility of data with normal distribution could not be confirmed with large probability according to the Shapiro–Wilk *W* test, data were also analyzed by non-parametric tests (Wilcoxon ranking test and the Friedman ANOVA test). However, these tests yielded exactly the same conclusions as parametric testing, supporting the robustness of the latter. For some data, the non-parametric Spearman’s correlation coefficient is shown. Statistical significance was defined as *p* < 0.05 in two-tailed testing.

## 3. Results and Discussion

### 3.1. Liver and Hindquarter Perfusion

During one-legged muscle stimulation, contraction force decreased rapidly (1272 ± 56 (mean ± SE), 446 ± 51, 431 ± 47, 429 ± 47 and 359 ± 27 g, respectively, at the beginning of the 1st, 5th, 10th, and 30th min and at the end of contractions). Hindquarter oxygen uptake increased 198 ± 44%, *n* = 6, from rest to the 15th min of contractions (*p* < 0.05) and then decreased again (28 ± 9%, *p* < 0.05) during the remaining stimulation period. Muscle lactate production peaked early and then decreased rapidly to basal levels during continued contractions (Figure 1). Glucose uptake in muscle increased gradually in the beginning of the contraction period and only decreased slowly after peak values had been reached (Figure 1). In response to a high physiological concentration of epinephrine (about 20 nM in cell-free perfusate) both lactate output and glucose uptake increased (*p* < 0.05, Figure 1), as found previously [29].

Initially, during muscle stimulation, hepatic glucose production increased (*p* < 0.05) to an extent comparable to the maximum increase in muscle glucose uptake (Figure 1). However, glucose production rapidly returned to basal levels not significantly different from zero (Figure 1). In response to epinephrine, glucose production showed a peak, which did not differ significantly from the early peak during contractions (*p* > 0.05, Figure 1). This in vitro hepatic response to epinephrine was about 20% of the maximum increases in hepatic glucose production previously seen in rats in vivo in response to stimulation by epinephrine, glucagon, or exercise [30]. Apparently, in vitro conditions are less favorable for hepatic glycogenolytic responsiveness than in vivo conditions. Hepatic glycogenolysis was not limited by the glycogen concentration, since this was 171 ± 13 μmol/g at the end of the experiment. From the onset of muscle stimulation, the liver began to take up lactate (Figure 1). The lactate uptake increased gradually to a plateau, which was maintained throughout contractions (Figure 1).

The fact that hepatic glucose production increased in response to muscle contractions to an extent comparable to the response to a high concentration of epinephrine is in agreement with the supposition that skeletal muscle may produce a hormone which enhances glycogenolysis in liver. The time course of hepatic glucose production agreed well with that of the contraction force, which, in turn, reflects the metabolic rate of the working muscle. On the other hand, glucose production did not vary in parallel with muscle glucose uptake. Compared to in vivo conditions, the fact that in the present in vitro experimental setup, liver and working muscles were connected in series should favor the disclosure of a liver active muscle hormone. Furthermore, the possible sites of biological hormone clearance were reduced and influence from the autonomic neuroendocrine system excluded in our in vitro system. On the other hand, in the present experiments, development of a marked increase in concentration of the putative hormone was restricted by hormone dilution in the necessarily large perfusate volume, by possible hormone adhesion to the tubes, and by the fact that the mass of contracting muscle producing the hormone was relatively small.

In order to involve as big a muscle mass as possible and to study the influence of the amount of active muscle, we also stimulated the sciatic nerves of both legs. All response patterns were similar to those seen during one-leg stimulation. However, total force production was 51% higher during two-legged compared to one-legged contractions. Hindquarter glucose uptake increased about 70% more during two- compared to one-legged contractions (Figure 2, *p* < 0.05), and lactate production (44%, Figure 2) and oxygen uptake (Appendix A, 40% late during contractions) were higher (*p* < 0.05) during the former condition. Furthermore, the increase in hepatic glucose output was higher (*p* < 0.05) and lasted longer, and hepatic lactate uptake was also higher (*p* < 0.05) during two- compared to one-legged contractions (Figure 1 and Figure 2). Muscle glycogen concentrations decreased markedly, but were not exhausted, during two-legged contractions (Appendix A).

The co-variation between active muscle mass and hepatic glucose production during contractions is compatible with release of a glycogenolytic hormone from the stimulated muscle. However, changes in other perfusate-borne factors might also account for the increase in hepatic glucose output. Perfusate flow was increased during contractions, and this might, for theoretical reasons, have resulted in increased glucose release. However, in separate experiments in which perfusate flow was varied, while hindquarters rested, no effect of the applied flow changes was found (Appendix A). Release of catecholamines from sympathetic nerves did not influence glucose output, as norepinephrine and epinephrine concentrations in hindquarter venous perfusate (epinephrine: 0.05 ± 0.02 n mol/L, (basal), 0.09 ± 0.02 and 0.07 ± 0.02 (5th and 55th min of contractions), norepinephrine: 0.14 ± 0.04 mol/L (basal), 0.13 ± 0.03 and 0.18 ± 0.03 (5th and 55th min of contractions) were lower than arterial concentrations in intact resting rats [31] and did not change in response to muscle contractions (*p* > 0.05).

Decreases in oxygen and carbon dioxide tensions, as well as an increase in pH, may enhance hepatic glycogenolysis. However, such changes did not occur in liver vein perfusate when muscle was stimulated (Appendix A). Oxygen and carbon dioxide tensions increased and pH decreased, but these changes were too small to influence hepatic glucose production [32], a conclusion confirmed by the finding of similar changes in control experiments without contractions and with increased flow (Appendix A), in which hepatic glucose output did not change (Appendix A). Excluding a direct regulatory role of the glucose concentration in the portal vein early during contractions, the decrease in this concentration was small and statistically not significant (Appendix A) and, furthermore, the concentration did not correlate with liver glucose output (r = 0.06, *p* > 0.05). Moreover, the subsequent gradual decline in portal glucose concentration with time (Appendix A) was accompanied by a decrease, and not by an increase, in glucose output (Figure 1 and Figure 2).

Lactate is a gluconeogenic substrate, and it is generally believed that the supply of lactate to the liver is important for hepatic glucose production in exercise [2,3,4,6]. In the present experiments, the time course of lactate release from muscle matched that of hepatic glucose output (Figure 1 and Figure 2). However, gluconeogenesis from lactate probably did not account for the contraction-induced rise in liver glucose production. This is so, as the time courses of hepatic uptake of lactate and output of glucose, respectively, did not correspond (Figure 1 and Figure 2): peak glucose output was significantly higher (*p* < 0.05, *n* = 13) than concomitant (and preceding) lactate uptake calculated in glucose equivalents. Furthermore, after its peak, the glucose output declined towards basal levels, while lactate uptake rose to a well-maintained plateau (Figure 1 and Figure 2).

The view that conversion of lactate to glucose did not explain the increment in liver glucose production during muscle stimulation was strongly supported by separate experiments on isolated livers. In these experiments, perfusate flow and lactate concentrations were varied to imitate the supply of lactate to the liver seen in the basal state and during contractions, respectively, in experiments with combined muscle and liver perfusion (Figure 3). During the liver perfusions, the increase in hepatic lactate supply was accompanied by a rapid increase in hepatic lactate uptake (Figure 3). Nevertheless, hepatic glucose output decreased steadily throughout and became zero at a time at which hepatic lactate uptake was identical to that seen in experiments with muscle stimulation (Figure 2 and Figure 3). The lactate taken up by the liver in the various in vitro experiments was probably predominantly oxidized [33]. The conclusion that, in the present experiments, gluconeogenesis from lactate did not contribute to hepatic glucose output agrees with estimates showing that hepatic glycogenolysis could well account for the measured glucose output. The estimates were based on liver glycogen concentrations measured at the end of the one-legged contraction experiment and from previous reports of hepatic glycogen concentrations and liver weight in short-term fasted rats.

### 3.2. Muscle Extract

The fact that, in experiments with combined muscle and liver perfusion, the effect of muscle contractions on hepatic glucose production was transient might reflect that force production and, accordingly, the perfusate concentration of a putative glycogenolytic muscle hormone waned rapidly, and that the isolated liver is not very sensitive to hormone action. Therefore, we also searched for stores of the putative hormone within muscle, and tested muscle extracts in intact rats. The same strategy applied to the heart resulted in discovery of atrial natriuretic peptide (ANP) [24,34]. Extracts of muscle were made using sodium phosphate buffer or boiling, the latter procedure denaturating proteins [25]. In response to intra-arterial injection in anesthetized rats of extract made by either procedure, a rapid and prolonged increase in plasma glucose concentration was seen (Appendix A). The concentrations of glucose (0.65 m mol/L) and lactate (2.3 m mol/L) in the extracts were far too small to account for the increase in plasma glucose concentration, and the extracts did not contain catecholamines (epinephrine and norepinephrine not detectable). Injection of an extract of epididymal fat tissue caused only a late and small increase in plasma glucose concentration, which was similar to that seen after NaCl injection (Appendix A).

These findings support the view that skeletal muscle contains a soluble non-protein factor which enhances hepatic glucose production. Evidently, the glucose-increasing effect of the muscle extract may not be physiologically relevant, but rather reflects an artifact caused by the extraction procedure. Furthermore, the effect of the extract may not be exerted directly on the liver, but might be indirect, e.g., secondary to a stress-hormone response. The effect of the extract may even reflect a reduction in glucose clearance rather than an increase in glucose production.

In order to clarify whether the extract in fact directly increased hepatic glucose production, its effect was studied in the perfused liver. Muscle extract added as a bolus increased hepatic glucose output (Figure 4). The time course was similar to that seen in experiments with perfused contracting muscle. Furthermore, the response was dose dependent (Figure 4) and probably due to glycogenolysis, as lactate output increased too (from 1.5 ± 0.3 μmol/min, *n* = 8, to 2.5 ± 0.1, *n* = 4, and 3.4 ± 0.6, *n* = 4, respectively, after 5 min exposure to two extract concentrations, which differed by a factor of three). The increase in glucose output after muscle-extract exposure was accompanied by an increase (*p* < 0.05) in liver oxygen uptake (49 ± 15%, *p* < 0.05, Appendix A), and also high hepatic venous oxygen tensions throughout perfusion (Appendix A) showed that the liver was not hypoxic. Catecholamines were undetectable in hepatic venous perfusate and muscle extracts.

Interleukin-6 has been shown to be a contraction-stimulated myokine [35], and it has been concluded that it contributes to hepatic glucose production in exercising man [36]. However, in the experiments underlying this view, subjects exercised at low intensity, while IL-6 was infused to reach a concentration corresponding to high-intensity exercise, and accordingly was unphysiological in the studied condition. Furthermore, other studies have shown that IL-6 deficiency in mice does not impair exercise-induced glucose production [37], and that IL-6 at physiologically relevant concentrations directly inhibits glucose production from the isolated rat liver exposed to plasma from exercising humans [12].

In order to characterize the glycogenolytic factor in our experiments, we carried out gel filtration of muscle extracts (Figure 5). Hepatic glucose production only rose in response to the late “Fraction V” (Figure 5). Livers responded within 1 min of exposure to the active fraction, and new responses were elicited upon repeated exposure (Figure 5). The fractionation confirmed that the active substance is different from IL-6, because its MW is below 2000 kDa, whereas that of circulating IL-6 varies between 22 and 27 kDa.

No other myokine than IL-6 has been associated with hepatic glucose production. Candidates might be low-molecular-weight peptides, such as ANP or ADH, as well as the small organic acid ß-aminoisobutyric acid (BAIBA) [38]. The latter is secreted from myocytes and plasma concentrations increase with exercise [38,39]. Furthermore, it has been shown to directly influence hepatic fat metabolism [38,40]. Even circulating adenosine and nucleotides may stimulate hepatic glycogenolysis [41]. This may involve purinergic signaling via G-protein-coupled receptors (GPCRs) activated by adenosine or nucleotides [42,43,44,45]. Speculatively, endocannabinoids might also influence hepatic glucose metabolism via GPCRs [46]. Undoubtedly, hitherto unknown exercise-responsive myokines will also be discovered through global mRNA sequencing and secretome and metabolome analysis [40].

In conclusion, based on an innovative, yet classical cross-circulation procedure, and analysis of muscle extract, the present study has provided evidence that contracting skeletal muscle may produce a hormone with a MW below 2000, which, by directly stimulating hepatic glycogenolysis, causes an increase in the supply of glucose to muscle that varies with the intensity of the contractile activity and, accordingly, with the energy needs. Such a hormone would fill up a gap in the present understanding of the regulation of hepatic glucose production in exercise [6,12]. It would also be in line with the finding that, during sleep, muscle inactivity and reduced metabolic rate and glucose uptake are accompanied by a fall in hepatic glucose output, which cannot be explained by known glucoregulatory mechanisms [47]. Still, before final acceptance of the new regulatory principle, the putative glycogenolytic muscle factor has to be further chemically characterized. Subsequently, its secretion from muscle during various exercise regimens, as well as its mechanism of action in the liver, should be explored.

## Figures and Tables

**Figure 1 jpm-12-00837-f001:**
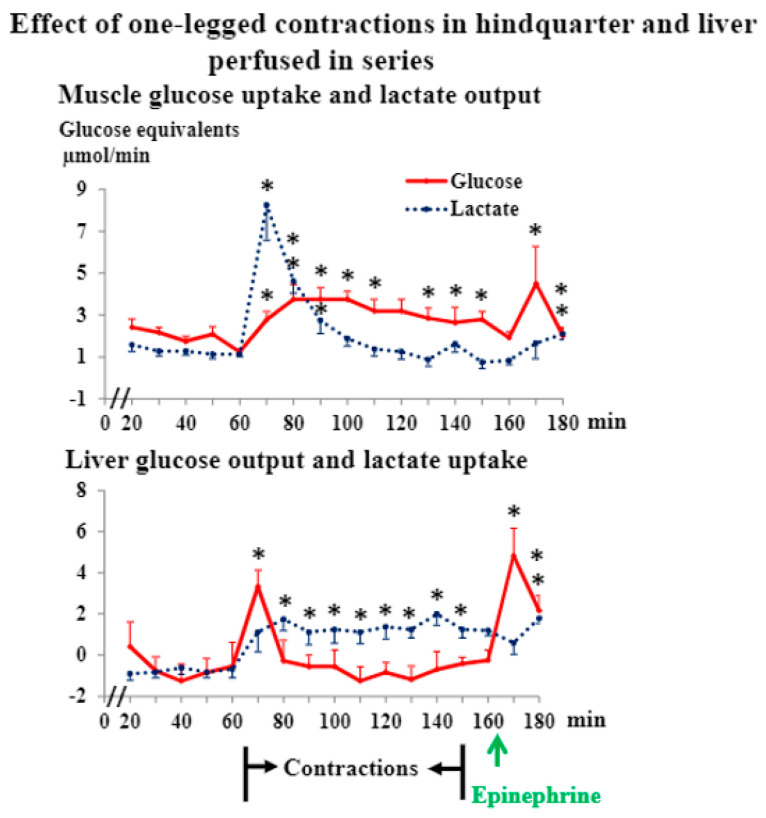
Data are means ± SE in 6 rats. * significantly different from basal level, *p* < 0.05.

**Figure 2 jpm-12-00837-f002:**
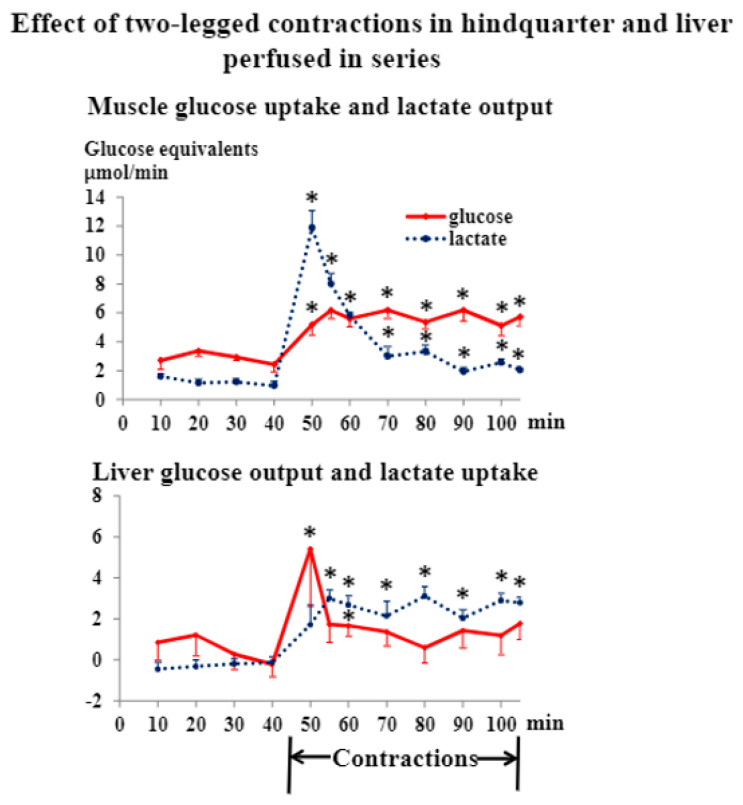
Data are means ± SE in 7 rats. * significantly different from basal level, *p* < 0.05.

**Figure 3 jpm-12-00837-f003:**
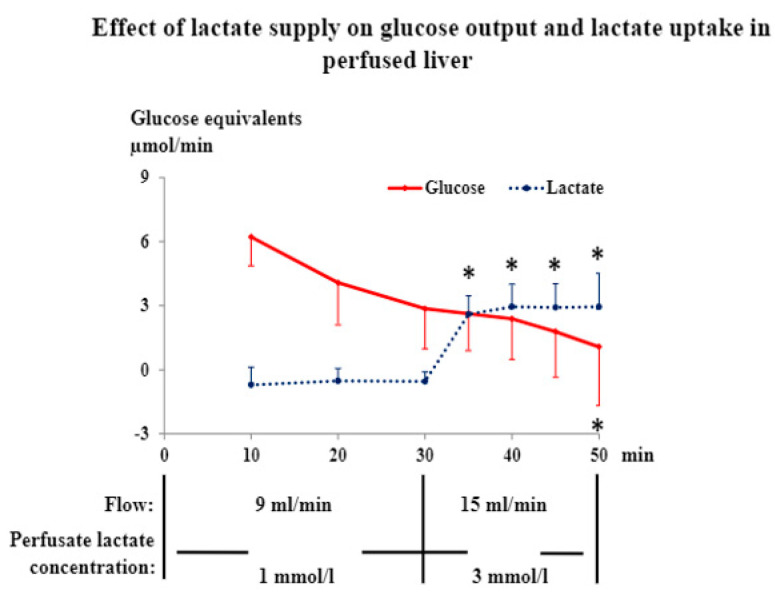
Data are means ± SE in 7 rats. * significantly different from the 30 min value, *p* < 0.05.

**Figure 4 jpm-12-00837-f004:**
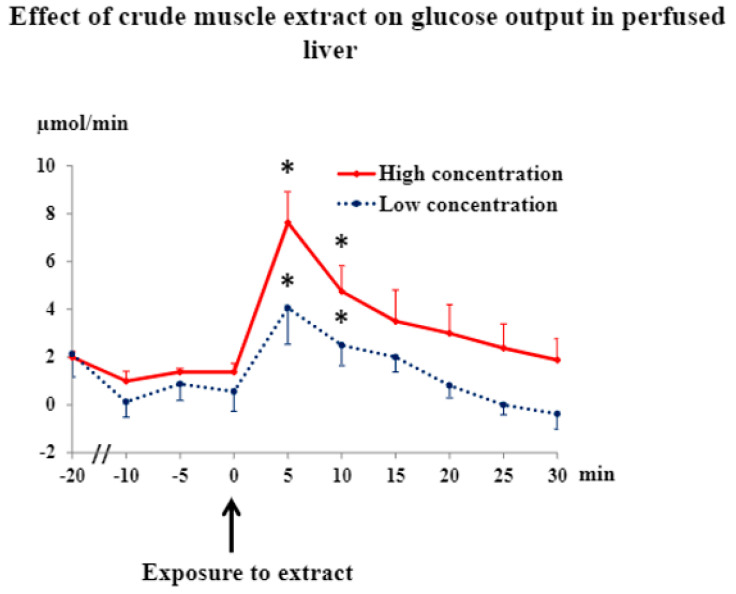
Hepatic glucose production was calculated from perfusate flow and portal and caval glucose concentration measurements. Values are means ± SE from 4 rats. * significantly different from basal values, *p* < 0.05.

**Figure 5 jpm-12-00837-f005:**
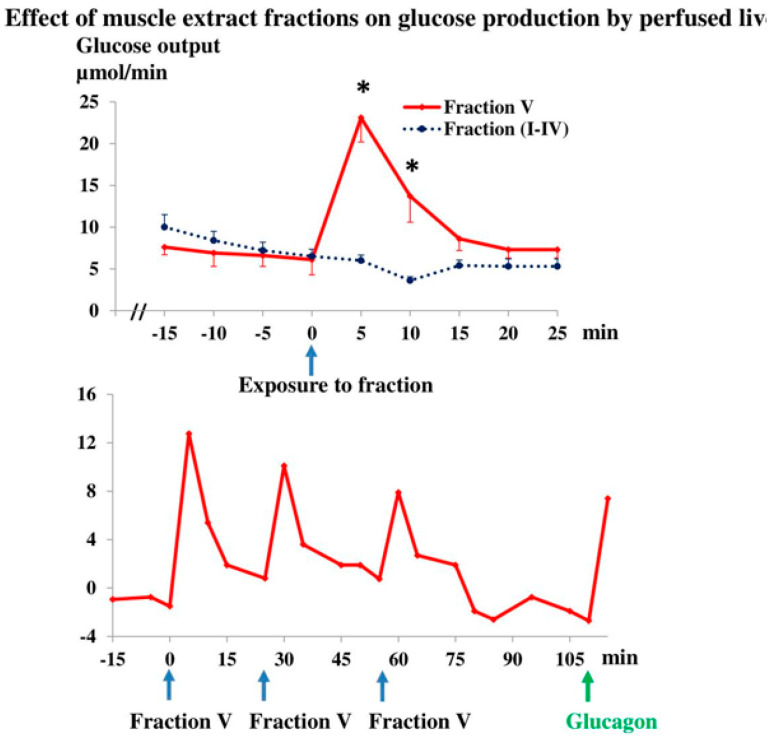
Upper panel: Values produced by 4–5 additions of each of fractions I–IV did not differ and were pooled accordingly. Five fraction V samples were studied. Values are means ± SE. * Fraction V value significantly different (*p* < 0.05) from basal values as well as from fraction I–IV values. Lower panel: Serial additions of three different fraction V samples and of glucagon, 10 nM, final concentration. Fractions were based on 7.2 g and 3.6 g muscle extract in upper and lower panel, respectively.

## Data Availability

Data are contained within the article or Appendix A and can on request also be made available from the corresponding author.

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
