# Peer review of "Studies in Rats of Combined Muscle and Liver Perfusion and of Muscle Extract Indicate That Contractions Release a Muscle Hormone Directly Enhancing Hepatic Glycogenolysis"

_jpm, 2022, doi:10.3390/jpm12050837_

Round 1
Reviewer 1 Report
Comment:
This paper discusses " Studies in rats of combined muscle and liver perfusion and of muscle extract indicate that contractions release a muscle hormone directly enhancing hepatic glycogenolysis. ". The main contribution of the paper is " it discussed whether contracting muscle produces a factor that directly stimulates hepatic glycogenolysis, using an innovative, yet classical cross-circulation procedure. The study provided evidence that contracting skeletal muscle may produce a hormone with a MW below 2000, which enhances hepatic glycogenolysis according to energy needs. "
This is an interesting study and is generally well written and structured. However, in my opinion the paper has some shortcomings in regards to signaling of GPCRs receptors and mechanism of these receptors with relation to glycogenolysis. Indeed, short paragraph about receptors in general (possibly cannabinoids and adenosine) and its relation to glycogenolysis or signaling in peripheral tissue is recommended to be added. This is important to highlight the significance of adenosine receptors. Moreover, cite more references are recommended.
In several instances I also suggested to cite more relevant and recent literature.
Important papers
Cannabinoids
- The Impact of CB1 Receptor on Inflammation in Skeletal Muscle Cells
https://pubmed.ncbi.nlm.nih.gov/34421307/
- The Impact of CB1 Receptor on Nuclear Receptors in Skeletal Muscle Cells
https://pubmed.ncbi.nlm.nih.gov/35366244/
Adenosine
1. Adenosine A2B Receptors - Mediated Induction of Interleukin-6 in Skeletal Muscle Cells
https://www.ncbi.nlm.nih.gov/pmc/articles/PMC7227993/
2. The Impact Of Adenosine A2B Receptors Modulation On Nuclear Receptors (NR4A) Gene Expression
https://biomedpharmajournal.org/vol9no1/the-impact-of-adenosine-a2b-receptors-modulation-on-nuclear-receptors-nr4a-gene-expression/
3. Adenosine Receptors Machinery and Purinergic Receptors in Rat Primary Skeletal Muscle Cells
https://biomedpharmajournal.org/vol7no2/adenosine-receptors-machinery-and-purinergic-receptors-in-rat-primary-skeletal-muscle-cells/
4. THE IMPACT OF ADENOSINE A2B RECEPTORS ON GLYCOLYSIS AND INSULIN RESISTANCE IN SKELETAL MUSCLE
https://www.researchgate.net/profile/Mansour-Haddad-3/publication/317828359_THE_IMPACT_OF_ADENOSINE_A2B_RECEPTORS_ON_GLYCOLYSIS_AND_INSULIN_RESISTANCE_IN_SKELETAL_MUSCLE/links/594d19df45851543382a6d05/THE-IMPACT-OF-ADENOSINE-A2B-RECEPTORS-ON-GLYCOLYSIS-AND-INSULIN-RESISTANCE-IN-SKELETAL-MUSCLE.pdf
5. THE IMPACT OF ADENOSINE A2B RECEPTORS MODULATION ON PEROXISOME PROLIFERATOR-ACTIVATED RECEPTOR GAMMA CO-ACTIVATOR 1-ALPHA AND TRANSCRIPTION FACTOR
https://www.researchgate.net/profile/Mansour-Haddad-3/publication/317828160_THE_IMPACT_OF_ADENOSINE_A2B_RECEPTORS_MODULATION_ON_PEROXISOME_PROLIFERATOR-ACTIVATED_RECEPTOR_GAMMA_CO-ACTIVATOR_1-ALPHA_AND_TRANSCRIPTION_FACTORS/links/594d165d0f7e9b49c70f8a82/THE-IMPACT-OF-ADENOSINE-A2B-RECEPTORS-MODULATION-ON-PEROXISOME-PROLIFERATOR-ACTIVATED-RECEPTOR-GAMMA-CO-ACTIVATOR-1-ALPHA-AND-TRANSCRIPTION-FACTORS.pdf
6. THE EFFECT OF NECA, CGS 21680, PSB 603 ON FATTY ACID TRANSPORT AND OXIDATION IN SKELETAL MUSCLE CELLS
https://www.researchgate.net/profile/Mansour-Haddad-3/publication/317828289_THE_EFFECT_OF_NECA_CGS_21680_PSB_603_ON_FATTY_ACID_TRANSPORT_AND_OXIDATION_IN_SKELETAL_MUSCLE_CELLS/links/594d1a0345851543382a6d09/THE-EFFECT-OF-NECA-CGS-21680-PSB-603-ON-FATTY-ACID-TRANSPORT-AND-OXIDATION-IN-SKELETAL-MUSCLE-CELLS.pdf
- Impact of Adenosine A2 Receptor Ligands on BCL2 Expression in Skeletal Muscle Cells
https://www.mdpi.com/2076-3417/11/5/2272
Minor comments:
- Well written except in some situations such as page 16 mimick ??. I advise recheck it again.
- The introduction should be advised to be re-written to be in more logical flow.
- I prefer to divide the results from discussion
- What about the GPCRs/ligands in this paper? (cannabinoids, adenosine,…)
- Cite more recent papers in peripheral tissue /inflammation. (above is suggested)
- How many animals used for each experiments?
- The methods in details should be described and analysis as well (analysis of glucose).
- What about cannabinoid/adenosine receptors in skeletal muscle?
- Please, Suggest future experiments in details
- Please, Specify the most specific protein from skeletal muscle that you suggest might be related lo liver glycogenolysis.
- What about skeletal muscle glucose uptake?
- What about fast/slow muscle? Which muscle you used?
- Expand in this paper the role of IL-6
- Please, try to add general paragraph about GPCRs cannabinids and adenosine receptors and discuss it importance to metabolism and inflammation in peripheral tissue such as skeletal muscle. (Above references are suggested)
- Please, discuss the role of NR4A in glycogenolysis
- Please, discuss the role of inflammation of peripheral tissue such as skeletal muscle and it relation glycogenolysis. (Short paragraph)
- Figure 1 and 2 are not obvious to me. It is not understandable.
- Although it needs to be in more logical flow, the introduction provides a good, generalized background of the topic. However, why not cite more literature papers (above).
- I think the motivations for this study need to be made clearer. In particular, the connection between skeletal muscle and glycogenlysis and inflammation.
- Regarding the figures: I recommend make more figures to be illustrative.
Given these shortcomings the manuscript requires Minor revisions.
"I recommend that this paper be accepted after minor revision."
Author Response
Thank you for having taken the time to carefully read our paper.
Regarding your main objections:
- “GPCR receptors and their involvement in glycogenolysis”.
We have now mentioned the possible involvement of GPCR receptors in hepatic glucose production and possible stimulation of these by nucleotides and even cannabinoids in p.21.
However, we have not extensively addressed the issue, because it is in our context quite speculative and beyond the scope of the paper, particularly as regards muscle.
- 21: “Even circulating adenosine and nucleotides may stimulate hepatic glycogenolysis [41]. This may involve purinergic signalling via G protein-coupled receptors (GPCRs) activated by adenosine or nucleotides [42-45]. Speculatively, also endocannabinoids might influence hepatic glucose metabolism via GPCRs [46].”
- “Cite more references”.
In fact, we thought that we had included rather many references (42) for an original article. However, we have now included further 5 references.
Please note, that we could unfortunately not on PubMed get access to most of the papers suggested by the reviewer. However, we think that those we have added will suffice.
…………………………………………..
Regarding Minor Comments:
- 16 “mimick” has been substituted by “imitate”.
The reviewer proposes to “re-write the Introduction”. However, earlier he stated that the text is well written and structured, and later that the Introduction provides a good, generalized background. So, we need more precise guidance to change the text.
“Divide the Results from Discussion”. In our Letter to the Editor we did emphasize that in accordance with the Guidelines of the JPM, we had used the possibility of including Results and Discussion in the same section. This was done to avoid considerable repetitions in the exposition. In fact, we think that combining Results and Discussion is an expedient way to present and explain a study, which could not be planned in detail in advance, but developed over time as new experiments had to be added.
“GPCRs, ligands; Cannabinoides, adenosine”. Have now been dealt with, see above.
“Cite recent papers in peripheral tissue/inflammation”. Our paper deals with the direct interaction between skeletal muscle and liver during attempted physiological conditions. Accordingly, we find that it is not within the scope of the paper to address other tissues and pathological conditions as e.g. inflammation.
“How many animals”. The number of animals included appears from the text in Materials and Methods and, in particular, from the legends to tables and figures.
“Analyses” are described in a paragraph on p.10 (also glucose and glycogen analyses).
“Receptors in skeletal muscle”. The scope of the study was to search for a factor released from muscle during contractions and acting on the liver. Accordingly, it would be far beyond the scope to deal with the many receptors existing in muscle.
“Future experiments in details”. We have extended the last paragraph (p. 21) to add further future experiments: “Subsequently, its secretion from muscle during various exercise regimens as well as its mechanism of action in the liver should be explored”.
“Suggest the protein most likely to be related to liver glycogenolysis”. In p. 19-21 we discuss the various factors which might account for the observed increase in glucose output from the liver during contractions. The conclusion is that the factor has a MW below 2000, a fact which excludes a multitude of proteins. However, we cannot tell whether the factor is at all a protein/peptide, and accordingly also not which protein/peptide would be the most likely candidate.
“What about muscle glucose uptake?”. Muscle glucose uptake responded to contractions (and epinephrine) as we and others have repeatedly seen (e.g. refs 4, 29). Glucose uptake and mechanisms for that were not the focus of the study. However, it was important to note that hepatic glucose production varied in parallel with contraction force and, accordingly, muscle metabolic rate, rather than with muscle glucose uptake (p. 11 and 12).
“Fast/slow muscle?” In p.7, last paragraph, the fiber type composition of the muscles used for glycogen analysis and a relevant reference are given.
“Expand the role of IL-6”. Because it has been thought that IL-6 contributes to hepatic glucose production in exercise, we discussed the possible role in a whole paragraph on p.19. The conclusion from re-analysis of earlier data and from the most recent data is that IL-6 plays no positive role. Furthermore, as pointed out on p. 20 our present fractionation analysis shows that the active substance released from muscle has a molecular weight well below that of IL-6. So, we think that IL-6 has been sufficiently dealt with in the paper.
“Add general paragraph about GCPRs in peripheral tissues and importance in metabolism and inflammation”. And “discuss inflammation of peripheral tissue”. These are very ambitious proposals, which, however, go far beyond the scope of the present paper, which deals only with metabolism in two tissues during attempted physiological conditions, and not with other tissues and inflammation. As described about we have now included information about GPCRs. And we have not been able to retrieve the proposed references.
”Figs. 1 and 2 difficult to understand” and “Make more figures”. We think that all information needed is in fact included in the heading, the subheadings, the text on axes, and the legend of Figs. 1 and 2. Furthermore, in the first draft we had put more information into the legend, but we shortened it, because the JPM´s guidelines advised that legends shall be brief and not repeat information, which can be found in e.g. Materials and Methods. However, we have now in the beginning of the Materials and Methods section added a short paragraph briefly explaining the experimental setup. This may also help understanding the figures. Also Figs S1 and S2 may help.
“Motivation needs to more clearly explain the connection between muscle, glycogenolysis and inflammation”. The aim was not to study pathophysiology and inflammation. That is the reason this is not mentioned in the Introduction.
Reviewer 2 Report
The abstract is much too long. The mixture of main and side experiments makes it seem very confusing and sometimes misleading.
The authors used a very complex model to investigate a complicated question. A drawing or diagram explaining the experimental setup would be helpful.
On the second paragraph on page 7, please do not write out the numbers.
Parts of the results such as “However, gluconeogenesis from lactate did probably not account for the contraction-induced rise in liver glucose production” (paragraph 3, page 17) belong in the discussion.
An explicit discussion is missing. Unfortunately, the paper is relatively unstructured, and there is no explicit separation between the results section and the discussion. This makes reading the paper considerably more difficult and makes it very confusing.
Author Response
- “Abstract is too long and main and side experiments are mixed”.
The Abstract contains only 266 words, which in general would not be considered much too many. Furthermore, the Guidelines of the JPM does not state an upper limit.
However, we have now reduced the Abstract by 10 words. Furthermore, we have avoided some of the mixing of main and side experiments.
- “A drawing explaining the experimental setup would be helpful”.
The time limit for the revision prevents provision of a drawing. Instead we have in the beginning of the Materials and Methods, p.5, added a short paragraph explaining the setup:
“A short explanation of the idea behind the experimental setup: Muscle (rat hindquarter) was perfused in vitro from the aorta, and the venous outflow served as perfusate for an isolated liver. Accordingly, factors released from muscle were carried by the perfusate directly to the liver. Muscle could be electrically stimulated and any effects of released factors on hepatic glucose output could be measured from perfusate flow and perfusate glucose concentration differences across the liver.”
- “P.7, second paragraph, do not write out the numbers”.
The writing of the numbers has been corrected.
- “Parts of the results belong in the discussion; an explicit discussion is missing”.
In our Letter to the Editor we did emphasize that in accordance with the Guidelines of the JPM, we had used the possibility of including Results and Discussion in the same section. This was done to avoid considerable repetitions in the exposition. In fact, we think that combining Results and Discussion is an expedient way to present and explain a study, which could not be planned in detail in advance, but developed over time as new experiments had to be added.